# TCGA Expression Analyses of 10 Carcinoma Types Reveal Clinically Significant Racial Differences

**DOI:** 10.3390/cancers15102695

**Published:** 2023-05-10

**Authors:** Brian Lei, Xinyin Jiang, Anjana Saxena

**Affiliations:** 1Krieger School of Arts and Sciences, Johns Hopkins University, Baltimore, MD 21218, USA; 2Biology Department, Brooklyn College, New York, NY 11210, USA; 3Department of Health and Nutrition Sciences, Brooklyn College, New York, NY 11210, USA; 4Biology and Biochemistry Programs, CUNY Graduate Center, New York, NY 10016, USA

**Keywords:** The Cancer Genome Atlas (TCGA), carcinoma, racial disparity, differential gene expression, race-based survival, gene-based survival, cancer pathway, DNA repair

## Abstract

**Simple Summary:**

Racial disparities in cancer incidence and outcome rates are prevalent in the US, with a variety of contributing factors, such as socioeconomic status, differences in lifestyle, environmental exposures, and biological and genetic determinants. The goal of this research was to broadly analyze public data to identify critical differences in molecular signatures and pathway regulation between races. Additionally, to support the clinical translatability of our work, we evaluated the association of differences in gene expression with patients’ survival outcomes. Our findings help inform the use of novel biomarkers in clinical settings and the future development of race-adjusted precision therapies.

**Abstract:**

Epidemiological studies reveal disparities in cancer incidence and outcome rates between racial groups in the United States. In our study, we investigated molecular differences between racial groups in 10 carcinoma types. We used publicly available data from The Cancer Genome Atlas to identify patterns of differential gene expression in tumor samples obtained from 4112 White, Black/African American, and Asian patients. We identified race-dependent expression of numerous genes whose mRNA transcript levels were significantly correlated with patients’ survival. Only a small subset of these genes was differentially expressed in multiple carcinomas, including genes involved in cell cycle progression such as *CCNB1*, *CCNE1*, *CCNE2*, and *FOXM1*. In contrast, most other genes, such as transcriptional factor *ETS1* and apoptotic gene *BAK1*, were differentially expressed and clinically significant only in specific cancer types. Our analyses also revealed race-dependent, cancer-specific regulation of biological pathways. Importantly, homology-directed repair and ERBB4-mediated nuclear signaling were both upregulated in Black samples compared to White samples in four carcinoma types. This large-scale pan-cancer study refines our understanding of the cancer health disparity and can help inform the use of novel biomarkers in clinical settings and the future development of precision therapies.

## 1. Introduction

Race is a very broad categorizer that groups populations according to common ancestral and phenotypic characteristics. Racial groups are social constructs that exhibit immense biological variation within them. Even so, previous work has identified genetic, epigenetic, and expression-level differences between racial groups that are implicated in a wide range of biological mechanisms [1]. Cancer epidemiologists have long recognized race-based differences in cancer incidence and outcomes, especially in the United States [2]. In addition to genetic variations linked to ancestry, external factors, such as culture, access to healthcare, socioeconomic status, and environmental exposures (e.g., pollution), have substantial impact on health disparity. Furthermore, studies suggest that self-reported race aligns closely with genetic ancestry and is a useful proxy for external determinants [2,3,4,5]. 

The debate is ongoing regarding the relative contribution of intrinsic and extrinsic factors in the cancer disparity. Some studies suggest that differences in poor outcomes cannot fully be explained by disparate access to care [6,7,8], while others suggest that no differences persist in mortality rates in equal-access healthcare environments or after adjustment for non-biological factors [9,10,11,12,13,14]. Such results are complicated by further stratifications along the lines of sex and age. For example, Lin et al. found no significant survival differences in renal cell carcinoma between White people and Black people after stratifying by age and sex [13]. On the other hand, Andaya et al. found that in colon cancer, survival differences between Black people and White people in an equal-access care system were evident only in patients younger than 50 years old [6].

American cancer registries have shown that five-year cancer survival rates are lower in Black people at each stage of diagnosis across a wide spectrum of cancer types [15]. The current published literature provides further insights into the cancer disparity in individual cancer types. For example, prostate cancer occurs more often and has greater mortality rates in African American men compared to Caucasian Americans, which has been increasingly linked to genetic and molecular alterations in addition to socioeconomic factors [16]. Likewise, incidence rates of aggressive endometrial cancers are significantly higher in non-Hispanic Black women compared to White women, and 5-year relative survival for Black women is significantly less than White people [17]. Islami et al. identified a major racial difference in liver cancer death rates that are linked to differences in risk factor prevalence, ranging from 5.5 per 100,000 in non-Hispanic White people to 11.9 per 100,000 in American Indians/Alaska Natives [18]. A racial examination of the California Cancer Registry revealed that advanced-stage and high-grade bladder cancer are especially prevalent in Black patients, along with significantly poorer 5-year disease-specific survival [19]. Tannenbaum et al. identified improved survival in Asian people compared to White people in Florida for non-small cell lung cancer [20].

Regardless of the relative importance of biological and non-biological factors, racial differences in incidence and survival are linked to many differentially expressed genes (DEGs). For example, Li et al. identified the *XKR9* gene, implicated in the exposure of phosphatidylserine during apoptosis, as being differentially expressed between Asian Americans, Caucasian Americans, and African Americans in many cancer types and significantly associated with overall survival. The authors suggest that *XKR9* could act as a potential race-dependent target for immunotherapy [21]. However, results from smaller-scale studies suggest that DEG patterns vary significantly across cancer types: Grunda et al. identified that the prognostically significant genes *AR*, *BCL2*, *CCND1*, *CDKN1A*, *CDKN1B*, *CDKN2A*, *ERBB2*, *ESR1*, *GATA3*, *IGFBP2*, *IL6ST*, *KRT19*, *MUC1*, *PGR*, and *SERPINE1* are differentially expressed in non-Hispanic White and African American breast cancer patients [22]. Vazquez et al. used The Cancer Genome Atlas (TCGA) data to discover that differential expression of chemokine receptors, when assessed with race and molecular subtype, could explain racial differences in the tumor microenvironment and response to immunotherapy [23]. Powell et al. highlighted race-dependent expression of genes, such as *AKT1*, *ALOX12*, *IL8*, *CXCR4*, *FASN*, and *TIMP3*, in prostate cancer, suggesting opportunities for targeted therapies [24]. Other prostate cancer research points to aberrant activation of signaling pathways, such as androgen receptor (AR), epidermal growth factor receptor (EGFR), and inflammation, across races as potential causes of racial difference in incidence and mortality [25]. Table 1 depicts a selection of DEGs between samples collected from White and Black patients that encompass a variety of cancer types and are implicated in many critical biological pathways. Additionally, other biological criteria, such as genetic polymorphism, mutational variation, and epigenetic variation, have also been linked to race-based differences in cancer incidence and outcome rates. For example, Asif et al. discovered CpG hypomethylation in Black patients with endometrial cancer compared to White patients indicated worse oncogenic transformation [26].

In this study, we aimed to analyze pre-existing datasets from The Cancer Genome Atlas (TCGA) with a focus on molecular differences between three races: White, Black, and Asian patients. We described the definitions of these race categories in detail in the Methodology section below. While poorer cancer prognoses and incidence rates have been associated with specific racial groups [15,31], the precise molecular mechanisms contributing to these differences remain the subject of ongoing research. Large transcriptomic and proteomic datasets available from a variety of studies and patient groups are often under-analyzed with respect to the racial component. In this study, we assessed these large datasets through the lens of race to gain insights about the racial dependency of DEGs, biological pathways, and patient survival. These revelations will be pivotal not only in providing cancer-specific sets of diagnostic and prognostic biomarkers but also in designing and developing race-adjusted precision therapies in the future.

## 2. Methodology

### 2.1. Data Information

We used TCGA datasets generated as part of the PanCancer Atlas tumor molecular analysis project, cataloged in Table 2. These datasets are already normalized to facilitate comparative analyses. We accessed and analyzed these data using the open-access, open-source cancer database cBioPortal [32,33,34]. The selected studies encompassed ten different cancer types, including the PanCancer Atlas datasets of colorectal adenocarcinoma (CORE, n = 338), uterine corpus endometrial carcinoma (UCEC, n = 445), invasive breast carcinoma (BRCA-P, n = 903), kidney renal clear cell carcinoma (KIRC, n = 383), kidney renal papillary cell carcinoma (KIRP, n = 253), liver hepatocellular carcinoma (LIHC, n = 329), lung adenocarcinoma (LUAD, n = 434), stomach adenocarcinoma (STAD, n = 344), and thyroid carcinoma (THCA, n = 370) [35]. In addition to these nine PanCancer Atlas datasets, we also included and analyzed two TCGA datasets profiled in 2015 *Cell* studies, namely, Invasive Breast Carcinoma (BRCA-C, n = 684) [36] and Prostate Adenocarcinoma (PRAD, n = 313) [37]. These two datasets also included gene methylation data. We chose both BRCA-C and BRCA-P datasets despite their significant overlap, as BRCA-P has nearly twice the number of Black patient samples compared to BRCA-C, evaluating reproducibility in our study. PRAD was the only study that did not include survival information associated with its samples. We assessed race-based survival with the samples that included all the information on race, mRNA expression, copy number alterations, protein expression and methylation; we filtered these data in the cBioPortal web tool.

In the TCGA database, each sample is assigned to only one of five racial categories, although in this study, we considered only three: “White”, “Black or African American”, and “Asian”. In this study, we use the term “Black” samples to refer to samples categorized as “Black or African American”. We excluded samples assigned to “American Indian or Alaska Native” and “Native Hawaiian or other Pacific Islander” due to low sample sizes, and samples for which race data were absent or unknown. The racial categories reported by TCGA are based on the definitions used by the US Census Bureau in classifying written responses to the race question [38,39]. Other than these self-identified categories, further information about the ancestry or racial admixture data are not available in TCGA. When discussing the literature, we have retained the racial designation as stated in corresponding publications.

### 2.2. Race-Based Kaplan–Meier Survival Analysis

We performed race-based survival analysis for each cancer type in the cBioPortal web tool using race as the comparison factor. For BRCA-C, BRCA-P, CORE, KIRC, KIRP, LUAD, and UCEC, we compared samples labeled as “White” against “Black”. For LIHC, STAD, and THCA, samples labeled “White” were compared against samples labeled “Asian”. In addition to these baseline racial analyses, we performed separate analyses after stratifying the racial groups by sex (e.g., White males vs. Black males, White females vs. Black females). Due to unequal levels of racial representation in the different cancer datasets, the racial groups that we selected for analyses varied. Survival was evaluated using four different survival metrics: disease-free survival, progression-free survival, disease-specific survival, and overall survival. We used log rank tests performed by cBioPortal to assess statistical significance; a *p*-value ≤ 0.05 was considered statistically significant.

### 2.3. Gene-Based Kaplan–Meier Survival Analysis

To determine the most clinically relevant genes for each cancer type, we used the Kaplan–Meier (KM) Plotter web tool [40]. The KM plotter database integrates expression data and clinical outcomes from the Gene Expression Omnibus (GEO), European Genome-phenome Archive (EGA), and TCGA [41]. We evaluated the prognostic value of the mRNA expression levels of a compiled set of 155 target oncogenes and tumor suppressor genes (see Appendix A) for all cancer types listed in Table 2, except for PRAD, for which these data were not available.

For each gene, samples were divided into two groups of low and high mRNA expression levels, based on the median expression, and the overall patients’ survival rates were compared. *p*-values and hazard ratios with 95% confidence intervals were calculated. To reduce the false discovery rate (FDR), we used *p* ≤ 0.01 as a threshold of statistical significance.

### 2.4. mRNA, Protein, and Methylation Analysis

In the cBioPortal web tool, we performed group comparison analyses with race as the comparison factor. For BRCA-C, BRCA-P, CORE, KIRC, KIRP, LUAD, PRAD, and UCEC, we compared “White” with “Black” samples, as reported. For LIHC, STAD, and THCA, “White” was compared against “Asian” samples. The racial groups selected for analysis varied due to unequal levels of racial representation in the different cancer datasets. For each study, results from mRNA and protein expression analyses (and DNA methylation analyses results for BRCA-C and PRAD) were downloaded as tab-separated values (TSV) files. We wrote and used a pandas-based Python (version 3.7.7) program to filter the data for the 155 target genes. Statistically significant differences in mRNA and protein expression between the two racial cohorts, as determined by the Benjamini–Hochberg procedure performed by the cBioPortal web tool, were identified.

### 2.5. Reactome Pathway Analysis

To perform integrative pathway analysis, we downloaded the PanCancer Atlas Reverse-phase protein arrays (RPPA) and clinical data from the original publication [42,43]. These data were not filtered for the 155 target genes for the purposes of thorough pathway analysis. We created a pandas-based Python program (https://github.com/brian-lei/cancer-racial-disparity, accessed on 18 January 2021) to format and export the data to Reactome, an open-source, open-access peer-reviewed pathway database [44,45]. We grouped samples by cancer type and by race, deleted duplicate samples, and removed data for a few selected proteins, namely, ADAR1, alpha-catenin, TTF-1, caspase-3, caspase-9, PARP1, and JAB1, with missing corresponding values. Post-translationally modified versions of proteins were also excluded from the analysis. We manually annotated samples according to their corresponding race in the Reactome web tool and performed a Pathway Analysis with Down-weighting of Overlapping Genes (PADOG) microarray analysis with race as the comparison factor.

### 2.6. Statistical Analysis as Presented in Results

Genes that received *p*-values less than or equal to 0.01 in gene-based KM survival analyses were considered “clinically significant”. The survival relationship, as evaluated by KM analysis, is denoted in subsequent tables with a “+” for positive and a “−“ for negative correlations between specific mRNA gene expression and overall survival. The associated log rank *p*-value is denoted in parentheses.

Genes that received q-values less than or equal to 0.05 in cBioPortal Benjamini-Hochberg comparison analysis were considered “differentially expressed” for mRNA expression, protein expression, or both. The race denoted in tables under “mRNA” is the racial group with *higher* mRNA expression of the gene, followed by the associated q-value in parentheses. The race denoted in tables under “Protein” is the race with *higher* protein expression and the associated q-value. “NS” represents non-significant values. Empty cells represent missing data from the original TCGA database.

Each table in the results section presents gene expression information for a particular cancer type. In all cases, we only included genes that were both clinically significant and differentially expressed between the denoted races. We excluded genes for which neither mRNA nor protein expression was significantly different.

In the Reactome Pathway Analysis subsection, differential regulation of pathways was considered significant with an adjusted *p*-value ≤ 0.05 and denoted in parentheses in each table header.

## 3. Results

The analyses are organized alphabetically by cancer type. Each section reports the results of cBioPortal differential expression analysis. All sections, except for PRAD, include cBioPortal race-based survival and KM gene-based survival analysis results. Gene methylation analyses were performed for only two studies for which data were available, BRCA-C and PRAD. Gene expression analysis was performed on only one dataset for each section/cancer type, except for Breast Invasive Carcinoma, which includes results from both BRCA-C and BRCA-P with significant overlap in samples. However, it is important to emphasize that even when BRCA-P included almost two-fold “Black” samples and an additional 132 “White” samples than BRCA-C, the overall analyses yielded similar results. Complete results and statistical significance on all 155 target genes are included in Appendix A. The original Reactome report with all queried pathways is included in Appendix A.

### 3.1. Breast Invasive Carcinoma (BRCA)

#### 3.1.1. Race-Based Survival 

In BRCA-C and BRCA-P, no significant differences were identified between White patients and Black patients for any of the survival metrics.

#### 3.1.2. Gene-Based Survival

High BCL3, FHIT, and IGF1R transcript levels were correlated with increased overall patient survival, while high CCNE1 and TGFBR1 transcript levels were correlated with decreased overall patient survival.

#### 3.1.3. Differential Expression

BRCA-C and BRCA-P data revealed a relatively high number of differentially expressed genes compared to the other studies. Most notably, in BRCA-C, CCNE1 mRNA and protein levels were significantly increased in Black samples compared to White samples. However, in BRCA-P, only the mRNA levels were significant, while protein levels remained non-significant (Table 3). Similarly, genes, such as APC, AXL, CDK4, CXCR1, ERCC4, IGF1, IGF1R, KRAS, MYB, PDGFD, RASSF8, and TGFBR1, had similar patterns of differential mRNA expression in both studies but lacked corresponding protein data. There were 87 other genes that exhibited differential expression across races; however, these were clinically non-significant, suggesting overall patient survival was not impacted. For example, BAK1, CCNB1, NOTCH1, RAD51, and STMN1 mRNA, and protein levels were enriched in Black samples in both studies, while KDR, MAPK9, and RAD50 mRNA, and protein were enriched in White samples in both studies.

#### 3.1.4. Methylation

In BRCA-C, CCNE1 methylation was significantly increased in White samples compared to Black samples. Thirty-eight other target genes exhibited differential methylation across racial groups, including tumor-suppressor genes, BRCA2 (higher methylation in White samples), and PTEN (higher methylation in Black samples).

### 3.2. Colorectal Adenocarcinoma (CORE)

#### 3.2.1. Race-Based Survival

In CORE, Black patients suffered from significantly worse progression-free survival (*p* = 0.0476) than White patients, while differences in disease-free (*p* = 0.443), disease-specific (*p* = 0.0513), and overall survival (*p* = 0.961) were not significant. While survival of Black vs. and White females did not differ significantly in any metric, Black males had significantly worse progression-free (*p* = 0.004526) and disease-specific survival (*p* = 0.0362) than White males.

#### 3.2.2. Gene-Based Survival

Out of all 155 target genes, only higher BIRC2 and FGF16 transcript levels were significantly associated with increased and decreased overall survival, respectively, in rectal adenocarcinoma. Expression of the remaining 153 genes did not impact overall survival and hence, were clinically non-significant.

#### 3.2.3. Differential Expression

While BIRC2 mRNA expression was not significantly different between White samples and Black samples, BIRC2 protein expression was significantly increased in Black samples (Table 4). Unfortunately, neither mRNA nor protein data were available for FGF16 in the original TCGA data, despite being clinically significant.

From the other 153 clinically non-significant genes, two genes, DIABLO and FHIT, differed in their expressions between Black and White samples. DIABLO mRNA expression was higher in White samples (protein levels were not significantly different), while FHIT expression was higher in Black samples (protein data not available). A total of 24 genes exhibited differential protein expression: BAK1, BIRC2, FASN, FOXM1, IRS1, MSH2, and STMN1 protein levels were significantly increased in Black samples, while CCNE2, CDH2, CDKN1A, CHEK1, DIRAS3, ERCC1, ETS1, GAPDH, HSPA1A, MAPK9, MYC, NF2, PRDX1, TGM2, TIGAR, VHL, and XRCC1 protein levels were significantly increased in White samples. The remaining clinically non-significant genes either had no data in TCGA or transcript levels did not differ between the examined races.

### 3.3. Kidney Renal Clear Cell Carcinoma (KIRC)

#### 3.3.1. Race-Based Survival

No significant differences were identified between White patients and Black patients, even after stratifying by sex, for any of the survival metrics.

#### 3.3.2. Gene-Based Survival

High APC, BCL2, CCND1, CDKN1B, CTNNB1, ERBB2, ERBB3, ERCC4, ERG, ETS1, FGF1, FGF12, FLT1, GAB2, IGF1R, IL6R, KDR, KRAS, MAPK1, MAPK9, MSH2, MYCN, NRAS, PDGFC, PDGFD, PTCH1, PTEN, RAD50, RASSF8, RB1, SCFD2, SMAD4, TAL2, TGFA, TGFBR2, and TGFBR3 transcript levels were correlated with increased overall patient survival.

High AXL, BAK1, BCL3, CCNB1, CCNE1, CCNE2, CRP, ERCC1, ETV4, FASN, FGF5, FGF8, FGF17, FGF21, FGF23, FOXM1, GAPDH, GNAS, GRP, IL6, NKX2-5, NKX3-1, NTRK1, SHBG, and SRC transcript levels were correlated with decreased overall patient survival.

#### 3.3.3. Differential Expression

AXL, FGF5, KRAS, NKX2-5, RB1, and TAL2 transcript levels were significantly higher in White samples compared to Black samples, although protein data were absent from the original TCGA data. NRAS transcript levels were also increased in White samples, but protein expression was not significantly different across races. ETV4 and FGF17 transcript levels were higher in Black samples, although protein data were absent. ERBB2 transcript levels were higher in Black samples, but protein did not differ significantly across races. ETS1, MAPK1, MAPK9, and RAD50 mRNA, and protein levels were significantly increased in White samples. BCL2, CCNE2, ERCC1, and GAPDH protein levels were also increased in White samples, although transcript levels were not significantly different across races. Interestingly, CCNB1, MSH2, FOXM1, and RAD51 protein levels were increased in Black samples, while transcript levels were increased in White samples (Table 5).

Other 37 genes exhibited differential expression across races but were clinically non-significant without any impact on overall survival. Interestingly, BAP1 transcript levels were increased in Black samples, while protein levels were higher in White samples. Several other genes had differential protein expression but not differential mRNA expression, namely, BECN1, BIRC2, CDKN1A, DIABLO, DIRAS3, IRS1, NF2, PRDX1, RAF1, STMN1, TGM2, TP53, VHL, and XRCC1.

### 3.4. Kidney Renal Papillary Cell Carcinoma (KIRP)

#### 3.4.1. Race-Based Survival

No significant differences were identified between White patients and Black patients, even after stratifying by sex, for any of the survival metrics.

#### 3.4.2. Gene-Based Survival

High AXL and SMAD4 transcript levels were correlated with increased overall patient survival, while high BRCA2, CCNE1, CCNE2, FASN, FGF5, FGF7, FGF8, FGF11, FGF18, FOXM1, GRP, IGF2, and PDGFRB transcript levels were correlated with decreased overall patient survival.

#### 3.4.3. Differential Expression

For all 155 target genes, including the genes identified by KM analysis as clinically significant for kidney renal papillary cell carcinoma, mRNA expression levels were not significantly different across races. Differences in protein levels were either not significant (e.g., BRCA2) or indeterminable due to a lack of associated protein data (e.g., AXL).

### 3.5. Liver Hepatocellular Carcinoma (LIHC)

#### 3.5.1. Race-Based Survival

No significant differences were identified between White patients and Asian patients, even after stratifying by sex, for any of the survival metrics.

#### 3.5.2. Gene-Based Survival

High BAK1, CCNB1, CCNE2, CDK4, CDKN2A, CHEK1, DIABLO, ETV1, ETV4, FOXM1, GAPDH, HRAS, MEN1, MSH2, NKX2-5, NRAS, PRDX1, RAD51, RASSF7, STMN1, TIGAR, and VHL transcript levels were all correlated with decreased overall patient survival. Interestingly, none of the target genes were correlated with improved survival.

#### 3.5.3. Differential Expression

In LIHC, protein data for the genes identified by KM analysis as clinically significant were either absent or displayed non-significant differences across races, except for BAK1, whose protein levels were significantly increased in White samples compared to Asian samples. BAK1 transcript levels, however, were not significantly different across races. CCNB1, CCNE2, CHEK1, FOXM1, HRAS, MSH2, RAD51, and STMN1 transcript levels were significantly higher in Asian samples, while TIGAR transcripts were increased in White samples (Table 6).

Other 31 genes exhibited differential expression across races but were clinically non-significant without any impact on overall survival; all these genes differed in mRNA expression only. Specifically, AKT1, ALK, BCL2, CDKN1A, CRP, EGFR, ERCC4, ETS1, FGF7, FLT1, GDNF, GRP, HBEGF, IGF1, IL6, IL6R, KIT, MDM2, NKX3-1, PDGFC, PDGFD, PTCH1, PTCH2, RASSF6, RET, TGFBR1, TGFBR2, and TGFBRAP1 mRNA levels were enriched in White samples, while CCNE1, GNAS, and MAPK9 mRNA levels were enriched in Asian samples. Protein expression data for these genes were either not significant or unavailable from the original TCGA database.

### 3.6. Lung Adenocarcinoma (LUAD)

#### 3.6.1. Race-Based Survival

No significant differences were identified between White patients and Black patients, even after stratifying by sex, for any of the survival metrics.

#### 3.6.2. Gene-Based Survival

High BCL2, FGF18, GAB2, KIT, NTRK1, PTCH1, RASSF2, and TMPRSS2 transcript levels were correlated with increased overall patients’ survival, while high CCNB1, CCNE2, FGF5, FGF19, FOXM1, GAPDH, IRS1, TGFA, and VEGFC transcript levels were correlated with decreased overall survival.

#### 3.6.3. Differential Expression

The mRNA transcript levels were not significantly different between White samples and Black samples for any of the genes identified by KM analysis as clinically significant for lung adenocarcinoma. Notably, IRS1 protein levels were significantly increased in Black samples (Table 7).

Seven clinically non-significant genes exhibited differential expression across races. ERCC1 and TIGAR protein expression was significantly increased in White samples, while NOTCH1 and RAD51 protein levels were increased in Black samples. mRNA levels for these four genes were not different between White samples and Black samples. MAPK1, MAPK9, and TGFBR1 transcript levels were enriched in White samples compared to Black samples; protein data were not significant for MAPK1/MAPK9 and absent for TGFBR1.

### 3.7. Prostate Adenocarcinoma (PRAD)

#### 3.7.1. Differential Expression

The mRNA transcript levels of CTNNB1, FGF19, NF2, PTCH1, and RB1 were elevated in White samples, while transcript levels of FGFR4 and RASSF7 were elevated in Black samples. Protein data for these genes were either not significant or missing. SMAD4 protein levels were increased in White samples, although mRNA levels were not significantly different between Black samples and White samples.

#### 3.7.2. Methylation

No significant methylation differences were identified in PRAD for any of the 155 examined genes.

### 3.8. Stomach Adenocarcinoma (STAD)

#### 3.8.1. Race-Based Survival

In STAD, Asian patients suffered from significantly worse disease-free survival (*p* = 0.0145) than White patients, while differences in progression-free (*p* = 0.729), disease-specific (*p* = 0.901), and overall survival (*p* = 0.304) were not significant. White females suffered from significantly worse overall survival compared to Asian females (*p* = 0.0453), while Asian males suffered from significantly worse disease-free survival compared to White males (*p* = 0.009962).

#### 3.8.2. Gene-Based Survival

High MSH2 transcript levels were correlated with increased overall patients’ survival, while high EGF, GRP, PDGFD, PDGFRB, TGFBR1, and VEGFC transcript levels were correlated with decreased overall survival.

#### 3.8.3. Differential Expression

The mRNA transcript levels were not significantly different between White samples and Asian samples for any of the genes identified by KM analysis as clinically significant for stomach adenocarcinoma. The GRP transcript data were absent from the database. Protein data were absent for all the genes that were identified as clinically significant, except for MSH2 protein, for which the data was not significantly different between White samples and Asian samples.

Two genes, PRDX1 and VHL, exhibited differential expression across races but were not clinically significant and did not impact overall survival. While PRDX1 transcript levels were not significantly different, protein levels were increased in White samples. VHL mRNA expression was increased in Asian samples, while protein levels were not different.

### 3.9. Thyroid Carcinoma (THCA)

#### 3.9.1. Race-Based Survival

No significant differences were identified between White patients and Asian patients, even after stratifying by sex, for any of the survival metrics.

#### 3.9.2. Gene-Based Survival

High EGF and FGF5 transcript levels were correlated with decreased overall patients’ survival.

#### 3.9.3. Differential Expression

Of the 155 target genes, none exhibited differential protein expression between White samples and Asian samples. EGF and FGF5 mRNA expressions did not differ significantly between White samples and Asian samples, and protein data for both genes were absent from THCA. The sole significant result was that TIGAR, a clinically non-significant gene, had enriched transcript levels in White samples.

### 3.10. Uterine Corpus Endometrial Carcinoma (UCEC)

#### 3.10.1. Race-Based Survival

In UCEC, Black patients suffered from significantly worse disease-specific survival (*p* = 0.0392), while differences in progression-free (*p* = 0.263), disease-free (*p* = 0.272), and overall survival (*p* = 0.331) were not significant.

#### 3.10.2. Gene-Based Survival

High FGF1, IGF1, and MDM2 transcript levels were correlated with increased overall patients’ survival, while high ALK, BAK1, CCNE1, CDKN2A, FGF11, FGF12, MSH2, and RET transcript levels were correlated with decreased overall survival.

#### 3.10.3. Differential Expression

MDM2 transcript levels were significantly higher in White samples, while FGF12 and CDKN2A transcripts were higher in Black samples; protein data were absent for these genes in the original TCGA data. CCNE1 mRNA and protein levels were both increased in Black samples compared to White samples. Additionally, while MSH2 mRNA levels were not significantly different across races, protein levels were elevated in Black samples (Table 8).

Other 30 genes exhibited differential expression across races but were clinically non-significant and did not affect overall survival. Specifically, ABL1, ETV1, FGF2, FGF5, GLI1, PDGFRA, PDGFRB, RASSF6, and TP53 transcript levels were elevated in White samples, while CASP12, CDH2, MYCL, and VEGFD transcript levels were elevated in Black samples. Protein data for these genes were either not significant or not available.

Additionally, BECN1, BIRC2, ERBB3, FASN, FOXM1, IRS1, NRAS, RAD51, and STMN1 protein levels were significantly increased in Black samples while CDKN1A, CTNNB1, ERCC1, ETS1, GAPDH, RAD50, and TGM2 protein levels were increased in White samples. However, mRNA levels were not significantly different across races for any of these genes. Interestingly, while NOTCH1 mRNA expression was significantly higher in White samples, protein levels were enriched in Black samples.

### 3.11. Reactome Pathway Analysis

We queried 987 pathways, of which 189 pathways were differentially regulated between races, at least in 1 cancer type. We observed that for 74 out of these 189 pathways, for each pathway, data for at least 3 proteins were available (see Appendix A). Importantly, homology-directed repair (HDR) was differentially regulated between Black and White samples in BRCA, CORE, KIRC, LUAD, and UCEC due to the differential protein expression of ten genes, namely, ATM, BRCA2, CHEK1, ERCC1, MRE11, PCNA, RAD50, RAD51, TP53BP1, and XRCC1 (Table 9). In CORE, HDR was overall downregulated in Black samples compared to White samples. In contrast, in four other cancer types, HDR was upregulated in Black samples indicating the existence of other cancer-specific differences.

In CORE, nucleotide excision repair was significantly downregulated in Black samples compared to White samples (adjusted *p* = 0.019) due to decreased XRCC1 and ERCC1 protein expression.

In LIHC, DNA mismatch repair pathways were upregulated in Asian samples compared to White samples due to elevated MSH6 protein levels (adjusted *p* = 0.024). Additionally, the G2/M DNA damage checkpoint pathway was upregulated in Asian samples in LIHC (adjusted *p* = 0.018). In Asian samples, ATM levels were higher while CDK1, TP53, YWHAB, and YWHAE levels were lower compared to White samples.

In BRCA, CORE, KIRC, and LUAD, ERBB4-mediated nuclear signaling was upregulated in Black samples compared to White samples due to the differential protein expression of six genes, namely, ESR1, PGR, SRC, STAT5A, STMN1, and YAP1 (Table 10).

In LUAD, TP53-mediated regulation of apoptotic gene transcription was increased in Black samples compared to White samples (adjusted *p* = 0.002): Black samples exhibited higher protein levels of ATM and BID, while TP53 and BAX expression was not significantly different between Black samples and White samples. In THCA, TP53 regulation of apoptotic gene transcription was increased in Asian samples compared to White samples (adjusted *p* = 0.017): Asian samples exhibited higher levels of BID protein. ATM, TP53, and BAX expression was also higher in Asian samples, although these differences were not significant.

In STAD, FOXO-mediated transcription of cell cycle genes was downregulated in Asian samples compared to White samples (adjusted *p* = 0.014).

## 4. Discussion

In this study, we dissected large transcriptomic, proteomic, and survival data available from The Cancer Genome Atlas (TCGA) to identify molecular differences between White, Black, and Asian samples that correlate with cancer health disparities. It is important to re-emphasize that TCGA data do not include detailed ancestry information, and hence, we have used the available racial categories instead. Although these racial categories are inherently subjective, others have emphasized the usefulness of race as it mostly aligns with genetic ancestry and can be a proxy to explain total variance that cannot be fully explained otherwise [3,4,5]. Thus, the findings in this study must not be confused with any intrinsic biological differences between populations. It is also worth mentioning that this study does not suggest any causal relationships; the underlying “cause-effect” relationships of much of the reported differential gene/protein expressions and survival disparities are unknown and must be addressed in further experimental and clinical setups. Nevertheless, these findings help illustrate the complex nature of the tumor molecular differences that exist between racial groups, particularly between White patients and Black patients. Importantly, there were insufficient data for extensive comparisons between White patients and Asian patients, highlighting the need for more proportional racial representation. However, our methods provide a valuable pipeline to analyze the race dependency of DEGs, proteins, and pathways and their survival association. We investigated the expression of 155 cancer-relevant genes and their negative or positive associations with patient survival to pinpoint race-dependent molecular factors in multiple cancer types.

We observed that most of the targeted genes were both differentially expressed and clinically significant in no more than one cancer type, except for a few that were significant in at least two cancer types; this is suggestive of cancer-specific variations in gene expression. For example, the cell cycle progression gene *CCNE1* was both differentially expressed and implicated in patient survival in breast and endometrial cancer. Additionally, the DNA repair gene MSH2 exhibited differential expression and clinical significance in kidney clear cell, liver, and endometrial cancer. *CCNB1*, *CCNE2*, and *FOXM1* were three other genes that were differentially expressed in more than one carcinoma with survival implications, specifically in kidney clear cell and liver carcinoma. The clinical implications of multiple members of the cyclin family of proteins and the cell cycle regulator *FOXM1* across multiple cancer types suggest that differences in cell cycle regulation are associated with survival disparities. Previous cancer-specific studies have also proposed this connection in breast [46] and endometrial carcinoma [30].

It is important to note that although we focused on genes that had a significant correlation with overall survival, we also identified many genes that had no impact on overall clinical survival yet were differentially expressed between the examined races. It may be useful to perform further research on this subset of genes and their relevant pathways to uncover any unapparent relationships, if any, between expression and patients’ survival.

Additionally, the source datasets varied in their abundance of differentially expressed biomarkers. While several pathways were identified in multiple cancer types as being regulated in a race-dependent manner with direct correlation to altered molecular patterns, in stomach adenocarcinoma, none of the clinically significant genes were differentially expressed between White and Asian samples.

A striking group of pathways that were significantly associated with race was DNA repair mechanisms. Most notably, racial differences between Black and White samples in the expression of genes pertaining to homology-directed repair, including *RAD50*, *RAD51*, *MRE11*, *ERCC1*, and *BRCA2*, were observed in five cancer types. Moreover, genes involved in nucleotide excision repair and DNA mismatch repair were differentially expressed in colorectal and liver cancer, respectively. Nuclear signaling mediated by ERBB4 was upregulated in Black samples in four cancer types, an important finding considering that overexpression of the kinase is associated with cancer development [47]. Table 11 lists the major findings of this study, indicating differentially regulated major pathways.

Protein data for many genes such as *CDKN2A* were absent from the original TCGA datasets; even so, our study corroborates the reported enrichment of *CDKN2A* transcript levels in Black patients with endometrial cancer [30]. We can also attempt to reconcile the mRNA differential expression analysis results with the existing literature on protein imbalances across racial groups. In UCEC, *MDM2* oncogene expression was higher in White samples, a result consistent with the reported elevation of MDM2 protein expression in Caucasian Americans when compared to African Americans in prostate cancer [48]. However, elevated transcript levels of *FGF12* in Black samples in UCEC conflict with the reported enrichment of fibroblast growth factor 12 in White patients [30]. We found many such contradictory results where transcript level differences did not accurately predict protein level differences (for example, MSH2, CCNB1, and FOXM1 in KIRC). Such inconsistencies between mRNA and protein differential expression may be attributable to post-transcriptional regulations of gene expression with racial impact; this is a promising topic for future research.

The autocrine-paracrine growth factor, insulin-like growth factor 1 (IGF-1), has been reported to be a promoter of cancer that inhibits the sex hormone-binding globulin (SHBG) and exists in elevated levels in African American individuals [1]. Interestingly, although they were not identified as clinically significant in breast cancer, cBioPortal analyses revealed significantly decreased levels of *IGF1* mRNA and elevated levels of *SHBG* mRNA in Black samples in both BRCA-P and BRCA-C studies.

This study provides a methodological pipeline to identify clinically valuable molecular targets with preexisting datasets. However, we recognize certain limitations. The use of broadly defined racial groups and cancer types in our analyses neglects their great variability and diversity (e.g., racially admixed people, different carcinoma subtypes, etc.). Genetic ancestry and tumor typing information would certainly be of interest to consider in future research. There is also a consistent lack of representation of Black and Asian patients in the data, an issue that must be adequately addressed by the broader research community. The source data only included protein information for a few hundred genes; hence our pathway analyses are not conclusive. We were unable to perform a comprehensive analysis of the prostate adenocarcinoma dataset as it lacked survival data. Finally, we were unable to perform multiple analyses of similar datasets, except BRCA-C and BRCA-P, as other datasets in TCGA and cBioPortal are predominantly represented by White patients and contain limited or no data for Black and Asian patients for adequate statistical comparison.

Nevertheless, in this study, we highlight correlative trends between patients’ survival and differentially expressed genes, suggesting that disparities in cancer outcomes overlap with key molecular differences. Future research should work with larger tumor datasets with a greater representation of Asian and Black patients to yield more statistically accurate analyses. Additionally, further work is needed to clarify the root causes of observed molecular differences to find direct relationships between differential expression and disparate outcomes. Ultimately, observed differences in cancer incidence and survival outcome are the result of a combination of many factors, including genetic polymorphisms, distinct molecular signatures, the synergistic effects of many traits, lifestyle, and environmental influences. Associations between self-reported race and disease outcome as presented in this study are thus valuable in predicting health impact of such structural inequalities along with genetic variations. With more proportional racial representation in tumor datasets and increased collection of proteomics data, we will be able to solidify our knowledge of molecular racial differences and work towards the development of personalized cancer therapeutics. Genome projects, such as the “1000 Genomes Project” [49] and the NIH-initiated “All of Us” [50], will provide more opportunities to identify unique molecular signatures in the future.

## 5. Conclusions

Racial patterns of differential gene expression and pathways’ regulation varied between carcinomas. The expression of certain cell cycle genes (*CCNB1*, *CCNE1*, *CCNE2*, and *FOXM1*) was race-dependent in several cancer types and correlated with significant survival differences. We observed racial variation in the regulation of several DNA repair mechanisms and oncogenic pathways. Importantly, homology-directed repair pathway was upregulated in Black samples compared to White samples in breast, endometrial, kidney renal clear cell, and lung carcinomas. ERBB4-mediated nuclear signaling, which is associated with cancer development, was upregulated in Black samples compared to White samples in breast, colorectal, kidney renal clear cell, and lung carcinomas. Mapping race-dependent differentially expressed genes that influence overall patients’ survival and illustrating their impact on proteomics and functional molecular pathways may provide valuable insights into targeting disparities in cancer.

## Figures and Tables

**Table 1 cancers-15-02695-t001:** Differentially expressed genes (DEGs) in breast, colorectal, endometrial, lung, and prostate cancers.

Cancer Type	Higher Expression in	DEGs	References
All	Black	IGF1, IL2RA, IL6	[1,21]
	White	XKR9, CST1, MTRNR2L1	
Breast	Black	CDKN2A, CRYBB2, SRC, SCN1A	[1,22,27]
	White	AR, BCL2, CASP8, CCND1, CDKN1A, CDKN1B, ERBB2, ESR1, GATA3, IGFBP2, IL6ST, KRT19, MUC1, PGR, SERPINE1	
Colorectal	Black	EGFR, TOP2A, CRYBB2, PSPH, IL33	[28,29]
	White	IL27, ADAL, ARSE, SMOC1, HESX1, TRNP1	
Endometrial	Black	MCM2, PLK1, MCM7, LAMA5, LAMC1	[30]
	White	SMC1B, CDC7, CCNE2, BRCA2, ERBB2, FGFR3	
Lung	White	CD274	[28]
Prostate	Black	AKT1, SPINK1, AR, ARA55, GNB3, POLR2L, TBP, EGFR, CRYBB2, IL6, IL8, CXCR4, FASN	[1,24,25]
	White	ALOX12, TIMP3	

**Table 2 cancers-15-02695-t002:** Breakdown of selected cancer datasets and respective sample counts. Gene expression analyses were performed either between White vs. Black or White vs. Asian samples, for a given cancer type.

Dataset Abbreviation	Cancer Type	Sample Count		
		White	Black	Asian
BRCA-C	Breast Invasive Carcinoma (Cell 2015)	594	90	
BRCA-P	Breast Invasive Carcinoma (PanCancer Atlas)	726	177	
CORE	Colorectal Adenocarcinoma	277	61	
KIRC	Kidney Renal Clear Cell Carcinoma	330	53	
KIRP	Kidney Renal Papillary Cell Carcinoma	193	60	
LUAD	Lung Adenocarcinoma	382	52	
PRAD	Prostate Adenocarcinoma	270	43	
UCEC	Uterine Corpus Endometrial Carcinoma	344	101	
LIHC	Liver Hepatocellular Carcinoma	174		155
STAD	Stomach Adenocarcinoma	258		86
THCA	Thyroid Carcinoma	320		50

**Table 3 cancers-15-02695-t003:** BRCA-C and BRCA-P, breast invasive carcinoma. Of all 155 target genes, 5 genes were identified by KM analysis as clinically significant (*p* ≤ 0.01). Of these, the four genes shown here were differentially expressed (q ≤ 0.05) between White vs. Black samples, in at least one of the two datasets.

Gene	Survival Relationship	BRCA-C mRNA	BRCA-C Protein	BRCA-P mRNA	BRCA-P Protein
	(*p*-value in parentheses)	(q-value in parentheses)			
BCL3	+(0.0063)	NS (0.127)		Black (0.002278)	
IGF1R	+(0.0026)	White (1.32 × 10^−4^)		White (3.27 × 10^−6^)	
CCNE1	−(0.0093)	Black (1.082 × 10^−5^)	Black (0.0087)	Black (2.22 × 10^−10^)	NS (0.577)
TGFBR1	−(0.0086)	White (0.0348)		White (3.081 × 10^−5^)	

**Table 4 cancers-15-02695-t004:** CORE, colorectal adenocarcinoma. Of all 155 target genes, 2 (BIRC2 and FGF16) were identified by KM analysis as clinically significant (*p* ≤ 0.01). Of these, only BIRC2 was differentially expressed (q ≤ 0.05) between White vs. Black samples.

Gene	Survival Relationship	mRNA	Protein
	(*p*-value in parentheses)	(q-value in parentheses)	
BIRC2	+(0.0017)	NS (0.605)	Black (0.004757)

**Table 5 cancers-15-02695-t005:** KIRC, kidney renal clear cell carcinoma. Of all 155 target genes, 61 genes were clinically significant in KM analysis (*p* ≤ 0.01); 21 genes were differentially expressed (q ≤ 0.05) between White vs. Black samples, as depicted below.

Gene	Survival Relationship	mRNA	Protein
	(*p*-value in parentheses)	(q-value in parentheses)	
BCL2	+(5.9 × 10^−6^)	NS (0.244)	White (0.008112)
ERBB2	+(1.7 × 10^−5^)	Black (0.007076)	NS (0.426)
ETS1	+(0.0002)	White (0.0322)	White (0.0285)
KRAS	+(0.0005)	White (0.0111)	
MAPK1	+(6.4 × 10^−6^)	White (5.349 × 10^−6^)	White (0.008112)
MAPK9	+(0.0003)	White (0.0002776)	White (0.0307)
MSH2	+(0.0011)	White (0.0353)	Black (0.0169)
NRAS	+(0.0073)	White (4.941 × 10^−5^)	NS (0.878)
RAD50	+(1.4 × 10^−6^)	White (0.004998)	White (3.4 × 10^−7^)
RB1	+(0.0001)	White (0.005863)	
TAL2	+(1.1 × 10^−8^)	White (0.0123)	
AXL	−(0.0083)	White (0.0113)	
CCNB1	−(0.0074)	White (0.0286)	Black (0.007517)
CCNE2	−(0.0006)	NS (0.1)	White (0.0259)
ERCC1	−(0.001)	NS (0.785)	White (0.000253)
ETV4	−(0.0062)	Black (0.0263)	
FGF5	−(6.8 × 10^−5^)	White (0.004835)	
FGF17	−(0.0028)	Black (0.0148)	
FOXM1	−(1.8 × 10^−5^)	White (0.006838)	Black (6.343 × 10^−5^)
GAPDH	−(0.0039)	NS (0.174)	White (5.163 × 10^−6^)
NKX2-5	−(1.5 × 10^−7^)	White (0.0372)	

**Table 6 cancers-15-02695-t006:** LIHC, liver hepatocellular carcinoma. Of all 155 target genes, 22 were identified by KM analysis as clinically significant (*p* ≤ 0.01). Of these, the following 10 were differentially expressed (q ≤ 0.05) between White vs. Asian samples.

Gene	Survival Relationship	mRNA	Protein
	(*p*-value in parentheses)	(q-value in parentheses)	
BAK1	−(0.008)	NS (0.366)	White (0.0446)
CCNB1	−(0.0008)	Asian (0.004704)	NS (0.176)
CCNE2	−(0.0011)	Asian (0.002332)	NS (0.22)
CHEK1	−(0.0002)	Asian (0.0256)	NS (0.22)
FOXM1	−(0.0015)	Asian (0.0238)	NS (0.374)
HRAS	−(0.0012)	Asian (0.0375)	
MSH2	−(0.0004)	Asian (0.00474)	NS (0.152)
RAD51	−(0.007)	Asian (0.003367)	NS (0.302)
STMN1	−(0.0004)	Asian (0.01)	NS (0.731)
TIGAR	−(0.0092)	White (0.002248)	NS (0.857)

**Table 7 cancers-15-02695-t007:** LUAD, lung adenocarcinoma. Of all 155 target genes, 17 were identified by KM analysis as clinically significant (*p* ≤ 0.01). Of these, only IRS1 was differentially expressed (q ≤ 0.05) between White vs. Black samples.

Gene	Survival Relationship	mRNA	Protein
	(*p*-value in parentheses)	(q-value in parentheses)	
IRS1	−(0.0099)	NS (0.669)	Black (0.0486)

**Table 8 cancers-15-02695-t008:** UCEC, uterine corpus endometrial carcinoma. Of all 155 target genes, 11 were identified by KM analysis as clinically significant (*p* ≤ 0.01). Of these, the following five were differentially expressed (q ≤ 0.05) between White vs. Black samples.

Gene	Survival Relationship	mRNA	Protein
	(*p*-value in parentheses)	(q-value in parentheses)	
MDM2	+(0.0013)	White (0.016)	
CCNE1	−(0.0001)	Black (0.0298)	Black (0.0146)
CDKN2A	−(9.7 × 10^−5^)	Black (0.0104)	
FGF12	−(0.0042)	Black (0.00292)	
MSH2	−(0.0081)	NS (0.106)	Black (0.0277)

**Table 9 cancers-15-02695-t009:** Differential expression of homology-directed repair (HDR) genes in CORE, UCEC, BRCA, KIRC, and LUAD. A yellow or blue highlighted header indicates overall HDR pathway upregulation in White or Black samples, respectively; the associated *p*-value is denoted in parentheses with the study abbreviation. Two yellow or blue dots indicate significantly higher protein expression in the corresponding race. A single-colored dot represents a non-significantly higher expression.

Protein	CORE (0.027)	UCEC (0.015)	BRCA (0.034)	KIRC (0.011)	LUAD (0.01)
ATM	●●	●	●●	●	●●
BRCA2	●	●●	●	●●	●
CHEK1	●●	●	●●	●	●
ERCC1	●●	●●	●●	●●	●
MRE11	●	●●	●●	●●	●
PCNA	●	●	●●	●	●
RAD50	●	●●	●●	●●	●
RAD51	●●	●●	●●	●●	●●
TP53BP1	●	●	●	●	●
XRCC1	●●	●	●	●●	●

**Table 10 cancers-15-02695-t010:** Differential expression in CORE, BRCA, KIRC, and LUAD of ERBB4-mediated nuclear signaling proteins. A blue highlighted header indicates overall signaling pathway upregulation in Black samples; the associated *p*-value is denoted in parentheses with the study abbreviation. Two yellow or blue dots indicate significantly higher protein expression in the corresponding race. A single-colored dot represents a non-significantly higher expression.

Protein	CORE (0.005)	BRCA (0.009)	KIRC (0.013)	LUAD (0.018)
ESR1	●●	●	●●	●
PGR	●●	●●	●●	●
SRC	●	●●	●	●
STAT5A	●	●●	●	●
STMN1	●●	●●	●●	●●
YAP1	●	●●	●	●

**Table 11 cancers-15-02695-t011:** Differentially expressed pathways and their respective carcinoma types. A: Asian, B: Black, W: White.

Pathway	Carcinoma Type (s)	Race with Higher Expression
DNA mismatch repair	Liver hepatocellular	A (vs. W)
ERBB4-mediated nuclear signaling	Breast, colorectal, kidney renal clear cell, lung	B (vs. W)
FOXO-mediated transcription of cell cycle genes	Stomach	W (vs. A)
G2/M DNA damage checkpoint	Liver hepatocellular	A (vs. W)
Homology-directed repair	Breast, endometrial, kidney renal clear cell, lung	B (vs. W)
	Colorectal	W (vs. B)
Nucleotide excision repair	Colorectal	W (vs. B)
TP53-mediated regulation of apoptotic gene transcription	Lung	B (vs. W)
	Thyroid	A (vs. W)

## Data Availability

The results shown in this study are in whole or part based upon data generated by the TCGA Research Network: https://www.cancer.gov/tcga (accessed on 18 July 2021). Data used in this article is publicly available and has been referenced. The computer code can be found at https://github.com/brian-lei/cancer-racial-disparity (accessed on 18 January 2021).

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
