# Peer review of "TCGA Expression Analyses of 10 Carcinoma Types Reveal Clinically Significant Racial Differences"

_cancers, 2023, doi:10.3390/cancers15102695_

Round 1
Reviewer 1 Report
Lei et al. report an analysis of clinical-molecular cancer databases by „race“, in an attempt to identify differences in molecular signatures or pathways which could either explain the different outcomes of patients depending on their “race”, as well as potentially identify “race”-specific molecular events and hence, therapeutic targets.
The word “disparity” is used incorrectly much of the time – whenever the authors are looking for differences that are kind of intrinsic – as well as the word “equitable” (l. 496) when they mean proportional.
Race is, as the authors describe, to a significant degree a social construct. In the USA, a person is considered (and may self-identify) as “black or African-American” if they have just one person of African descent in the family, whereas in Brasil a person is considered white if they have just one Caucasian somewhere in the family tree. The same applies to “Asian”. How are people of mixed ethnicity identified and how are their data treated?
Selection of a single dataset per tumor type is problematic. Observations that are biologically defined (non-random) will be reproduced across studies, lack of reproducibility across studies is demonstrated by the included breast cancer studies and also by contradictory conclusions for HDR by race depending on cancer type, thus quite possibly the reported inter-racial differences may not be reproducible. Reproducibility should be explicitly queried before the authors’ conclusions are supported. Since different cancer subtypes are associated with very different outcomes, the lack of refinement of the analyses with respect to clinical entity is likely to distort the analysis.
What is non-Hispanic Black (l. 62), what is the difference between Black and African-American (l. 425)? The manuscript is overall poor on definitions. The Legends are not self-explanatory.
Author Response
"Please see the attachment."

Reviewer 2 Report
The presented work is focused on the comparison of carcinoma molecular profiles between racial groups. The authors analyzed publicly available data to identify patterns of differential gene expression in tumors obtained from White, Black/African American, and Asian patients. The findings reveal a small subset differentially expressed in multiple carcinomas. The results suggest that race-dependent differences in transcriptional signatures have an impact on biological pathway regulation in cancer.
The manuscript is well-written and contains a valuable summary of the obtained results. However, the critical problem is related to the definition of race factor that was used in this study. The analyses were restricted to the comparisons between groups defined as White vs. Black/African American groups and White vs. Asian. The is no information about the ethnicity or ancestry of the individuals. The approach was extremely simplified in the most important part of the study. Moreover, significant differences in transcripts abundance levels were detected but it is not possible to tell whether the alterations are causes or effects of various predispositions or etiology of particular carcinoma types between the races.
It is recommended to use other sources of data to analyze the cancer-related molecular differences between the races in more detail.
Author Response
"Please see the attachment."

Round 2
Reviewer 1 Report
The manuscript is much improved. I have no specific comments. The vague definition of "black" which does not acknowledge potential effects of mixed ethnicity but instead treats "blackness" as a dominant trait strongly reduces the potential contribution of the work to the field of race-adjusted medicine.
Reviewer 2 Report
Additional data was not included in the revised version of the manuscript. The comparison of cancer-related gene expression profiles between the races is very limited. In the present form, the results are too preliminary to be published in the Cancers journal.
Round 3
Reviewer 2 Report
No further comments.